

# Tumour-associated macrophages mediate the invasion and metastasis of bladder cancer cells through CXCL8

Hao Wu, Xiangxiang Zhang, Dali Han, Jinlong Cao and Junqiang Tian

Department of Urology, Lanzhou University Second Hospital, Lanzhou, Gansu, P.R. China
Urology Institute, Lanzhou University Second Hospital, Lanzhou, Gansu, P.R. China
Key Laboratory of Gansu Province for Urological Diseases, Lanzhou, Gansu, P.R. China

Corresponding author
Junqiang Tian, ery_tianjq@lzu.edu.cn

## ABSTRACT

Tumour-associated macrophages (TAMs) are associated with both the progression and poor prognosis of a variety of solid tumours. This study aimed to investigate and clarify the tumour-promoting role of CXCL8 secreted by TAMs in the urothelial carcinoma microenvironment of the bladder. Immunohistochemistry ($n = 55$) was used to detect Chemokine (C-X-C motif) ligand 8 (CXCL8), CD163 (a TAM marker), Matrixmetalloproteinase-9 (MMP-9), vascular endothelial growth factor (VEGF), and E-cadherin in cancerous and adjacent tissues of bladder cancer patients. TAMs-like PBM (peripheral blood mononuclear)-derived macrophages were developed using *in vitro* experiments. T24, 5637, and UM-UC-3 were treated with conditioned medium (CM) for the experimental intervention group, without CM for the blank control group, and with CM and an anti-CXCL8 neutralizing antibody for the experimental control group, respectively. The immunohistochemical study showed that the expression of CXCL8 was significantly upregulated as the number of infiltrating TAMs increased in the tumour tissues. A high expression of CXCL8 significantly correlated with an increase in the expression of MMP-9 and VEGF and a decrease in expression of E-cadherin in the microenvironment. This revealed that TAM-derived CXCL8 is highly associated with bladder cancer migration, invasion, and angiogenesis. The concentration of CXCL8 was significantly higher in CM collected from TAM-like PBM-derived macrophages than that from THP-1 cells. In subsequent in vitro experiments, we found that CM derived from TAM-like PBM-derived macrophages can also increase the migration rate, invasiveness, and pro-angiogenic properties of tumour cells. Additionally, the effect of CXCL8 was significantly diminished by the addition of an anti-CXCL8 neutralizing antibody to CM. The infiltration of TAMs in the tumour microenvironment leads to the elevation of CXCL8, which in turn promotes the secretion of MMP-9, VEGF, and E-cadherin by bladder cancer cells. This alters the migration, invasion, and pro-angiogenic capacity of bladder cancer cells and accelerates cancer progression.

## INTRODUCTION

Tumorigenesis is a highly complex process involving multiple factors and specific subsets of macrophages have been found to be indeed involved in tumorigenesis. (*Gordon et al., 2017*). In the tumour microenvironment, macrophages with the M2 phenotype polarized by IL-4, IL-10, etc., are often referred to as "tumour-associated macrophages" (TAMs). TAMs are a major component in leukocyte infiltration in tumour tissues (*Hao et al., 2012*; *Noy & Pollard, 2014*). CD163 (a hemoglobin scavenger receptor) is a unique TAM marker (*Almatroodi et al., 2016*; *Barros et al., 2013*). After recruiting tumour-derived chemokines (mainly Chemokine (C-C motif) ligand 2 (CCL2) and colony-stimulating factor 1 (CSF-1) (*Noy & Pollard, 2014*), TAMs are often confined to the perivascular region or the tumour-infiltrating margin. Upon arrival, they can secrete pro-migratory factors (e.g., epidermal growth factor, or EGF), degrade extracellular matrix proteolytic enzymes, accelerate tumour migration, and induce tumour invasion and metastasis (*Huang & Feng, 2013*; *Pollard, 2004*; *Quail & Joyce, 2013*). It has been shown that in solid tumours, including bladder cancer, an increased number of CD163[+] TAM infiltrates is positively correlated with a poor clinical outcome (*Komohara, Jinushi & Takeya, 2014*; *Lewis & Pollard, 2006*; *Wang et al., 2015*).

Solid tumours and their surrounding areas are composed of inflammatory cells as well as various types of chemokines and cytokines (*Stadler et al., 2015*). CXCL8 is considered a prototypic chemokine belonging to the CXC family, which is responsible for the recruitment and activation of neutrophils and granulocytes to inflammation sites (*Waugh & Wilson, 2008*). CXCL8's role in cancer is widely recognized. It not only participates in tumour angiogenesis, but also activates MMP to promote metastasis-related tissue remodeling (*Azenshtein et al., 2005*; *Kim et al., 2001*). The expression of CXCL8 in urine specimens and the tumour microenvironment of bladder cancer patients is significantly higher than that of healthy volunteers (*Pignot et al., 2009*; *Sheryka et al., 2003*), and the overexpression of CXCL8 is positively correlated with significantly reduced survival of patients overall (*Zhang et al., 2014*). It has been shown that TAMs in the solid tumour microenvironment can secrete CXCL8 in large amounts, directly contributing to the progression of pancreatic ductal carcinoma, papillary thyroid cancer, colorectal cell carcinoma, and esophageal squamous cell carcinoma (*Chen et al., 2018*; *Fang et al., 2014*; *Hosono et al., 2017*; *Lee et al., 2012*).

However, little is known about the role of CXCL8 derived from TAMs in the para-epithelial carcinoma network of the human urinary tract in the bladder tumour microenvironment. In this study, we focused on CXCL8 secreted by TAM-like PBM-derived macrophages, and determined the critical role and biological effects of TAM-derived CXCL8 on bladder cancer cell production.

## MATERIALS AND METHODS

### Patient selection and data collection

The Second Hospital of Lanzhou University approved this research, and all participants provided informed consent for this study. Immunohistochemical staining was performed on 55 pathologically diagnosed bladder cancer patients who underwent total or partial

cystectomy between September 2017 and December 2018. All hematoxylin and eosin (H&E)-stained specimens obtained by initial cystectomy were reassessed independently by experienced uropathologists.

## Immunohistochemical staining and quantification

Paraffin-embedded sections for immunohistochemical detection were first deparaffinized in xylene and rehydrated in ethanol, and added to a pressure cooker with 0.01 M citrate (pH = 6.0) buffer and antigen recovery. After pressure was reached, the sections were incubated in the pressure cooker for 2 min. Endogenous peroxidase activity was blocked using 1% hydrogen peroxide in methanol according to the Streptavidin-Peroxidase Detection Kit (ZSGB-BIO, China) after blocking in 1% bovine serum albumin (BSA) for 10 min. Slides were incubated overnight at 4 °C with anti-human CXCL8 antibody (Abcam, CA, goat polyclonal, diluted 1/1,000 in 1% BSA), anti-CD163 antibody, anti-VEGF antibody, anti-MMP-9 antibody, and anti-E-cadherin antibody (Santa Cruz, CA, goat polyclonal, diluted 1/250 in 1% BSA), respectively. Slides were then incubated with biotinylated anti-goat IgG secondary antibody for 10 min at room temperature. Sections were subsequently stained with 3, 3'-diaminobenzidine (DAB; Vector Laboratories), counterstained with hematoxylin (Solarbio, Peking, China), dehydrated, and coverslipped. The results were observed under a microscope (OLYMPUS BH-2 Microscopes, USA), and the positive staining rate was counted.

The semi-quantitative methods used for immunohistochemistry results were as follows: CD163-positive round cells were counted in cancerous areas using at least three independent high-power microscopic fields (HPF; 400×, 0.0625 $\mu m^2$). The mean number of CD163-positive cells was considered the number of tumour-infiltrating TAMs. To quantify the expression levels of CXCL8, MMP9, VEGF, and E-cadherin in bladder cancer cells, tumour cells were counted in at least three independent fields (HPF; 400×, 0.0625 $\mu m^2$), and that number was divided by the total cancer cells to calculate the percentage of positive cells (1–100%). The evaluation was carried out blindly by two investigators, without knowledge of the patients' outcome or other clinic pathological characteristics.

## Cell lines and reagents

Three bladder cancer cell lines, 5637, UM-UC-3, and T24, as well as acute monocytic leukemia cell line THP-1 and EA·hy926 (Cell Bank of the Chinese Academy of Sciences, Peking, China) were used in the present study. Cell lines were maintained in RPMI-1640 medium (Gibco, Thermo Fisher Scientific, USA) supplemented with 10% fetal bovine serum (Pan Biotech, Germany), 100 units/mL penicillin, and 100$\mu$g/mL streptomycin in a standard humidified incubator at 37 °C in 5% $CO_2$. Phorbol-12-myristate-13-acetate (PMA) was purchased from Multis Sciences (LIANKE) Biotech (Hangzhou, China). Recombinant protein IL-4 (PeproTech, USA) was also used in the described experiments.

## Generation of TAMs from THP-1

To generate TAMs from monocytic THP-1, cells were seeded $1 \times 10^6$ cells in a dish and treated with PMA (200nM) for 24 h for differentiation into resting macrophages (M0 cells). For M2 polarization, cells were treated with IL-4 (20 ng/mL) for an additional 48 h

**Table 1  qRT-PCR primers (human).**

| | | |
|---|---|---|
| **CXCL8** | Forward | **CAAGCTGGCCGTGGCTCT** |
| | Reverse | **TGGGGTGGAAAGGTTTGGAGT** |
| **CD163** | Forward | **CCGGGAGATGAATTCTTGCCT** |
| | Reverse | **GGCCTCCTTTTCCATTCCAGAAA** |
| **MMP-9** | Forward | **GTACTCGACCTGTACCAGCG** |
| | Reverse | **TTCAGGGCGAGGACCATAGA** |
| **VEGF** | Forward | **GAGTACCCTGATGAGATCGAGT** |
| | Reverse | **ATTTGTTGTGCTGTAGGAAGCT** |
| **E-cadherin** | Forward | **ATGGCTGAAGGTGACAGAGC** |
| | Reverse | **CACCTTCCATGACAGACCCC** |
| **GADPH** | Forward | **CAGGA GGCAT TGCTG ATGAT** |
| | Reverse | **GAAGG CTGGG GCTCA TTT** |

(*Genin et al., 2015*; *Mantovani et al., 2004*). qRT-PCR analysis for CD163 was performed to confirm TAM generation.

## Enzyme-linked immunosorbent assay (ELISA)

A human CXCL8 kit and MMP-9 kit were purchased from Neobioscience Technology Company, China, and ELISA was performed according to the manufacturer's instructions. Both standard and tissue samples were added to a 96-well plate with 0.1 mL of biotinylated antibody and incubated at 37 °C for 60 min. After gentle washing, enzyme conjugate working solution was added to each well and the samples incubated for 30 min. Chromogenic substrate (TMB) was then added and the samples incubated for 15 min in the dark. After stopping with stop solution, the optical density at 450 nm was quantified using a microplate reader (Bio Tek Instruments, USA) for further analysis.

## Quantitative real-time polymerase chain reaction (qRT-PCR)

RNA was extracted from cells using TRIzol (Takara, Japan) as per the manufacturer's instructions. Conversion to cDNA was achieved using cDNA PrimeScript[TM] RT Master Mix (Perfect Real Time) (Takara, Japan) in a reverse transcription PCR instrument (Bio-Rad Laboratories, USA). Quantitative qRT–PCR was carried out using the CFX Real-Time PCR System (Bio-Rad Laboratories, USA) in a 15-µl reaction volume containing first-strand cDNA, TB Green[TM] Premix Ex Taq[TM] II (Tli RNaseH Pluse). Primer sets can be found in Table 1. Relative fold changes in mRNA levels were calculated after normalization to GADPH using the comparative Ct method.

## Western blot analysis

After the treated cells were collected and washed with PBS and RIPA buffer lysis, they were placed on ice for 30 min. The concentration of supernatant protein was determined using the BCA method (Solarbio, Peking, China) after centrifugation at a speed of 14,000 r.p.m. for 15 min. The same amount of total protein was resolved using SDS-PAGE and transferred to the PVDF membrane. Antibodies to the target protein were tested overnight at 4 °C. The primary antibodies used in this study were anti-E-cadherin (dilution,1:100),

anti-VEGF (dilution,1:200), and anti-actin antibody (dilution, 1:1000; ZSGB-BIO, China) as an internal loading control. After incubation with IRDye® near-infrared fluorescence secondary antibody, the protein was detected using Odyssey Near-Infrared Fluorescence Imaging Systems (LI-COR Biosciences, USA).

## Cell invasion assay using conditioned media obtained from TAMs

THP-1-derived TAMs were seeded in serum-containing media onto cell culture dishes at a density of 1,000,000 cells/dish. Twenty-four hours later, the medium was replaced with serum-free RPMI1640 medium, and cells were incubated for an additional 48 h. The CM was collected from manipulated TAMs, followed by centrifugation to remove cells or cell debris. The invasion assay was performed using Costar® Transwell® inserts (Corning Incorporated, USA) according to the manufacturer's directions, and 24-well plates with 8-$\mu$m pore membranes were lightly coated with 2 mg/mL Matrigel (Solarbio, Peking, China). T24, 5637, and UM-UC-3 cells were added to each insert at a density of 30,000 cells/well in the CM described above. The lower chamber contained RPMI1640 medium with 10% FBS as a chemoattractant. The cells were maintained in a humidified incubator with 5% $CO_2$ at 37 °C for 48 h. Invading cells on the membrane were then fixed using paraformaldehyde, stained with 0.1% crystal violet (Solarbio, Peking, China), and positively stained cells were counted. The cell numbers counted from five random fields were averaged.

## Wound healing assay

Cells were seeded in six-well plates and incubated overnight. A vertical bar was drawn using a mark pen on the back of the six-well plate, and a 1mL pipette tip was used to transversely scratch the cell monolayer. After washing the scraped cells from the wells with PBS, serum-free medium, CM, and CM with added anti-CXCL8 neutralizing antibody were added to the wells. Tumour cell migration was observed at 0, 6, 12 and 24 h. Wound healing percentage was calculated using Image J software.

## Vascular endothelial cell tube formation experiment

Matrigel (8.7 mg/mL) was placed in a refrigerator at 4 °C to slowly melt overnight, and 50$\mu$L was spread in each well of a 96-well plate and placed in an incubator at 37 °C for 1 h. The EA·hy926 cells were prepared into a cell suspension of $2 \times 10^5$ cells/mL, which was mixed well and added to a 96-well plate at a volume of 100 $\mu$L per well. The CM prepared in advance was then added at a volume of 50 $\mu$L per well, incubated in an incubator at 37 °C and 5% $CO_2$, and observed after 8 h (*Arnaoutova & Kleinman, 2010*; *Lee & Kang, 2018*; *Ma et al., 2017*). We observed the tube formation of endothelial cells and counted the number of vessels and visible nodes.

## Statistical analysis

All experimental results were expressed as mean $\pm$ SD and analyzed for statistical significance by a two-tailed Student's $t$-test. The interrelationship was examined using Spearman's analysis. Data were statistically analyzed and plotted using IBM SPSS version 23 (SPSS Inc., Chicago, IL, USA) and Prism software version 5 (San Diego, CA, USA), respectively. A $p$-value $< 0.05$ was considered statistically significant.

## RESULTS

### The association between the expression of CXCL8 in bladder cancer and clinicopathological features

The clinicopathological characteristics of 55 bladder cancer cases are outlined in Table 2. Representative pictures of immunohistochemical staining for CXCL8, CD163, MMP9, VEGF, and E-cadherin are shown in Figs. 1A–1J. Since CXCL8 is mainly expressed in the cytoplasm of tumour cells, the expression levels were significantly higher in bladder cancer tissues than in normal tissues (Fig. 1K). Strong CXCL8 expression was observed in the tumor tissue of high-grade tumors, while low-grade tumors had relatively lower expression levels. The expression of CD163 (a marker of TAMs), MMP9, VEGF, and E-cadherin was quantitatively assessed and compared with the CXCL8 expression score. The results showed that a high expression of CXCL8 in tumour tissues was associated with a high infiltration of TAMs, high expression of VEGF and MMP9, and a low expression of E-cadherin (Figs. 1L–1O). When compared to clinicopathological variables, a higher CXCL8 expression score was associated with higher T category and tumor grade (Table 2). These findings suggest that the infiltration of TAMs and the expression of CXCL8 in bladder cancer exhibit the aggressiveness of the remodeled stroma.

### TAM-like PBM-derived macrophages can secrete CXCL8

To demonstrate how macrophages in the microenvironment can secrete CXCL8, we used TAM-like PBM-derived macrophages with IL-4-induced M2 phenotype (Figs. 2A–2C). The TAM-like PBM-derived macrophages were cultured in a serum-free medium, which was collected daily. The amount of CXCL8 secreted in the collected culture medium was then measured using an ELISA experimental method. It can be concluded that different concentrations of IL-4 *in vitro* induce different amounts of CXCL8 secreted by TAM-like PBM-derived macrophages (Fig. 2D). The highest amount of CXCL8 was secreted when the concentration of IL-4 was 20 ng/mL (Fig. 2E). After analyzing the expression of CXCL8 and CD163 using qRT-PCR, we concluded that CXCL8 expression significantly increased (Fig. 2F) and TAM-like PBM-derived macrophages induced by IL-4 significantly expressed the specific M2 macrophage marker CD163 (Fig. 2G).

### CXCL8 derived from TAM-like PBM-derived macrophages promotes the invasion of bladder cancer

To examine the potential consequences of recruiting macrophages into bladder cancer tumour microenvironments, we investigated the potential effect of TAM-derived CXCL8 on bladder cancer cell invasion. We tested invasion ability after culturing bladder cancer cells 5637, T24, and UM-UC-3 with/without CM, using 8 μm membrane inserts as described in the Materials and Methods. The results show that CM increases bladder cancer cells' invasion capacity, and this effect can be suppressed using an anti-CXCL8 neutralizing antibody (Figs. 3A–3J). RNA subsequently extracted from bladder cancer cells after the CM treatment described above showed that the expression of MMP-9 had significantly increased in the treated cells (Fig. 3K). This was also demonstrated by ELISA
**Table 2   Clinicopathologic characteristics of 55 cases with bladder cancer.**

| Variables | Cases | CXCL8 expression in tumor Number of cases (percentage) | | P |
| --- | --- | --- | --- | --- |
| | | Low | High | |
| Total | 55 | 18 | 37 | |
| Age | | | | 0.78 |
| <65 | 29 | 9 (30%) | 20 (70%) | |
| ≥65 | 26 | 9 (35%) | 17 (65%) | |
| Sex | | | | 0.31 |
| Male | 44 | 13 (30%) | 31 (70%) | |
| Female | 11 | 5 (45%) | 6 (55%) | |
| Tumour size | | | | 0.41 |
| Less than 3 cm | 9 | 4 (44%) | 5 (56%) | |
| 3 cm and more | 46 | 14 (30%) | 32 (70%) | |
| Tumour stage | | | | 0.01* |
| <T2 | 13 | 9 (69%) | 4 (31%) | |
| ≥T2 | 42 | 9 (21%) | 33 (79%) | |
| TumoUr grade | | | | 0.02* |
| Low grade | 6 | 4 (65%) | 2 (35%) | |
| High grade | 49 | 11 (22%) | 36 (78%) | |
| Lymph node metastasis | | | | 0.13 |
| Negative | 23 | 10 (44%) | 13 (56%) | |
| Positive | 33 | 8 (25%) | 25 (75%) | |
| Vascular invasion | | | | 0.73 |
| No | 38 | 13 (42%) | 25 (66%) | |
| Yes | 17 | 5 (29%) | 12 (71%) | |
| Distant metastasis | | | | 0.93 |
| No | 31 | 10 (33%) | 21 (67%) | |
| Yes | 24 | 8 (31%) | 16 (69%) | |

$P$ value was analyzed by a chi-square test; * indicates $P < 0.05$ with statistical significance.

at the protein level (Fig. 3L), indicating that CXCL8 can promote bladder cancer invasion by stimulating bladder cancer cells to secrete MMP-9.

## CXCL8 derived from TAM-like PBM-derived macrophages increases the migration of bladder cancer cells

We conducted a wound healing experiment to investigate the effect of CXCL8 on bladder cancer cell migration. We planted 5637, T24, and UM-UC-3 bladder cancer cells on a six-well plate, attached them to the wall for 24 h, and then added CM (Figs. 4A–4R). We found that the cell migration increased significantly after the addition of CM. After neutralizing CXCL8 in CM with anti-CXCL8 neutralizing antibody, the accelerated migration rate was observed to be significantly reduced. This suggests that CXCL8 can promote the migration of cancer cells (Fig. 4S).

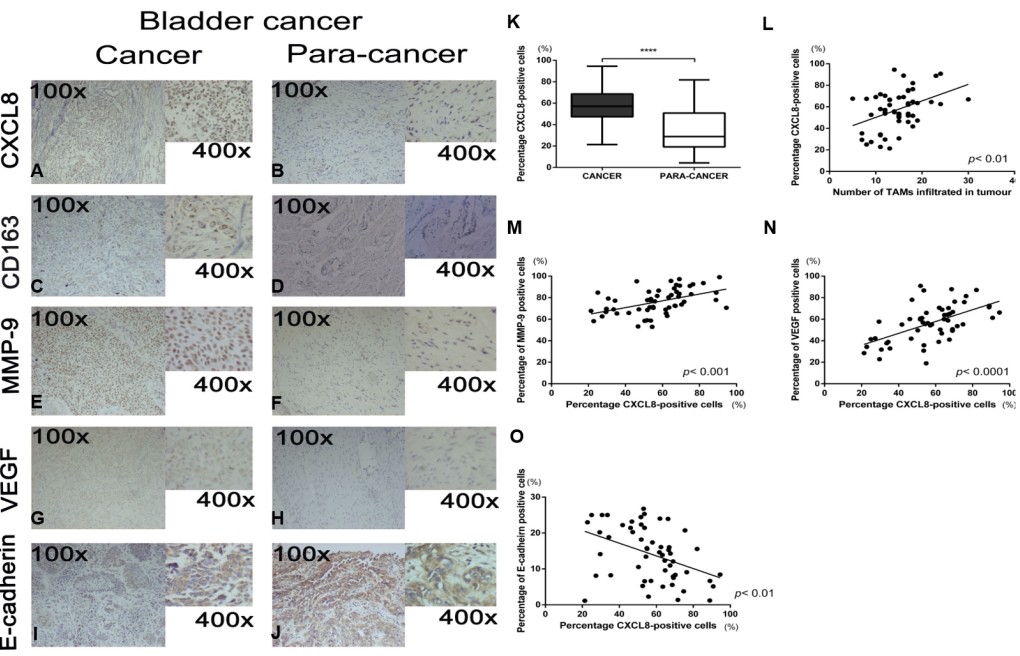

**Figure 1** **Recruitment of tumor-associatedmacrophages and CXCL8 expression in human bladder cancer, and the correlation between the expression of CXCL8 and the expression of MMP-9, VEGF, and E-cadherin.** (A–J) Representative expression status of CXCL8, CD163, MMP-9, VEGF and E-cadherin in human bladder cancer and para-cancertissues. All images were captured at $100\times$ and $400\times$ magnification. (K) Expression of CXCL8 in bladder cancer and para-cancertissues. In the box-and-whisker plot, significance was assessed by the paired student's $t$ test. ****$p < 0.0001$. (L–O) The interrelationship between the percentage of CXCL8 positive cancer cells and (L) the number of infiltrated TAM, (M) percentage of MMP-9 positive cells, (N) percentage of VEGF positive cells, and (O) percentage of E-cadherin positive cells using Spearman's correlation. Spearman $r$ was found to be 0.41, 0.47, 0.57, −0.42, respectively. (95% confidence interval, 0.16–0.61, 0.24–0.66, 0.36–0.73, −0.61−−0.17, respectively).

## CXCL8 derived from TAM-like PBM-derived macrophages promotes the formation of blood vessels in tumours

Angiogenesis is associated with the benign-malignant transition of tumours, with resulting blood vessels providing tumour cell nutrients and oxygen for their propagation, invasion, and metastasis. The blood vessel formation process is complex, and TAMs are one of the major contributors (*Lin & Pollard, 2007*). To investigate the proangiogenic ability of CXCL8 derived from TAM-like PBM-derived macrophages, we used the serum-free RPMI-1640 and CM described above to culture T24 cells for 24 h, and extracted the culture medium, which was centrifuged to become new CM (called CM-C and CM-T24, respectively). Using the endothelial cell tube formation assay described above, we added CM-C (as control) and CM-T24 as EA·hy926 cells were added to the 96-well plate. After 8 h of culture, we observed significantly more endothelial cell tube formation in wells with CM-T24 added than in those with CM-C. We used CM to culture T24 cells with an additional anti-CXCL8 neutralizing antibody, and then cultured EA·hy926 cells in the extraction medium (CM-T24-NA) described above. We found that the tube formation of EA·hy926 cells was significantly inhibited (Figs. 5A–5D). Total RNA and total protein

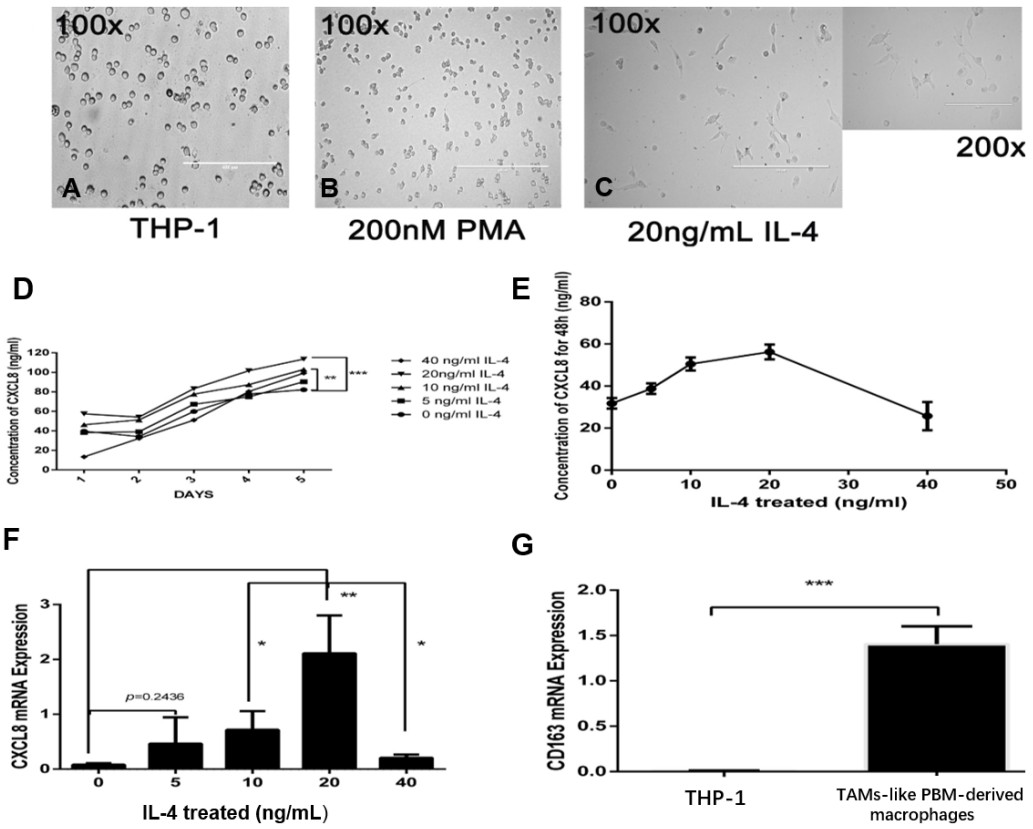

**Figure 2** **IL-4-induced TAM-like PBM-derived macrophages can secrete CXCL8.** (A–C) Morphological changes of THP-1. PMA (200 nM, 24 h) and IL-4 (20 ng/mL, 48 h) sequentially induce the formation of TAMs-like PBM-derived macrophages. Prior to treatment, THP-1 cells were round, floating, and did not attach to the bottom surface of the flask; after treatment with PMA and IL-4, they adhered to the bottom surface with abundant cytoplasmic processes. (D) After M0 induction using different concentrations of IL-4, the concentration of CXCL8 in serum-free medium was measured over time by ELISA. **$p <$ 0.01, ***$p <$ 0.001. (E) When the concentration of IL-4 was 20 ng/mL, the amount of CXCL8 secreted by TAMs-like PBM-derived macrophages was the greatest. (F) CXCL8 mRNA expression level was determined by qRT-PCR. (G) The expression level of CD163 in TAM-like PBM-derived macrophages. Expression levels are expressed as the mean ± SD. *$p < 0.05$, **$p < 0.01$, ***$p < 0.001$.

were then extracted from the CM-treated bladder cancer cells. QRT-PCR and WB analysis showed that the expression level of VEGF in CM-treated bladder cancer cells increased, indicating that CM can induce angiogenesis by promoting the secretion of VEGF in bladder cancer cells. The inhibition of CXCL8 with the CXCL8-neutralizing antibody reduced the ability of CXCL8 to promote VEGF secretion in bladder cancer cells, suggesting that CXCL8 in CM plays a role in this process (Figs. 5E–5G).

## DISCUSSION

It has been well-established that the microenvironment of solid tumors is constituted of malignant cells and some non-malignant mesenchymal cells (*Balkwill & Mantovani, 2001*; *Nilendu et al., 2018*). Among these non-malignant cells, macrophages play an important

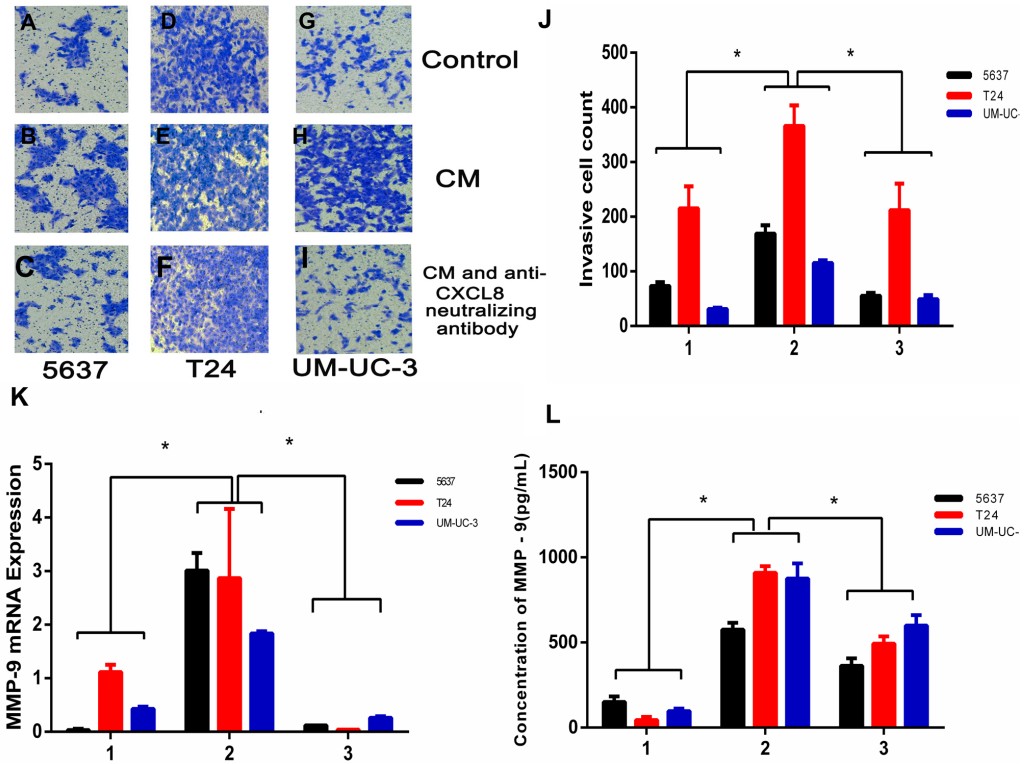

**Figure 3** **TAM-like PBM-derived macrophages increase the invasiveness of bladder cancer cells through CXCL8.** (A–I) For Transwell invasion assays, 5637, T24, and UM-UC-3 cells were seeded at $5.0 \times 10^5$ cells/well on a Matrigel-coated Transwell chamber in serum-free RPMI-1640. Serum-free RPMI-1640, CM, and CM with anti-CXCL8 neutralizing antibody were added to the Transwell chamber. The cell insert was placed on a 24-well plate containing RPMI-1640 with 10% FBS for 48 h. The invasive cells on the membrane were stained and counted. (J) The number of bladder cancer invasive cells on the membrane (1, Control; 2, CM; 3, CM with anti-CXCL8 neutralizing antibody). The results were expressed as the mean $\pm$ SD. *$p < 0.05$. (K–L) The expression level of MMP-9 in 5637, T24, UM-UC-3 was determined by (K) qRT-PCR and (L) ELISA (1, Control; 2, CM; 3, CM with anti-CXCL8 neutralizing antibody). Expression levels are expressed as the mean $\pm$ SD. *$p < 0.05$.

role in promoting tumour migration, invasion, and new blood vessel formation. TAMs in the tumour microenvironment can lead to pro-tumorigenic inflammation, which plays an important role in tumorigenesis (*Mantovani & Sica, 2010*), but TAMs can also express both chemokines and cytokines to promote an immunosuppressive tumour microenvironment (*Guo et al., 2016*). The infiltration of TAMs in primary tumours is associated with a poorer prognosis across almost all tumours (*Mantovani et al., 2017*), and when macrophage infiltration in the tumour microenvironment is blocked using a colony stimulating factor-1 (CSF-1) or colony stimulating factor-1 receptor (CSF-1R) inhibitor, the tumour burden in patients can be significantly reduced (*Ries et al., 2014*). Although it has been established that TAMs are associated with the progression of solid tumours and poor patient prognoses, the specific TAM mechanism in tumour cells has not been fully explained. This study found that the infiltration of TAMs in bladder cancer patients significantly increased when compared with the normal control, and TAM levels positively

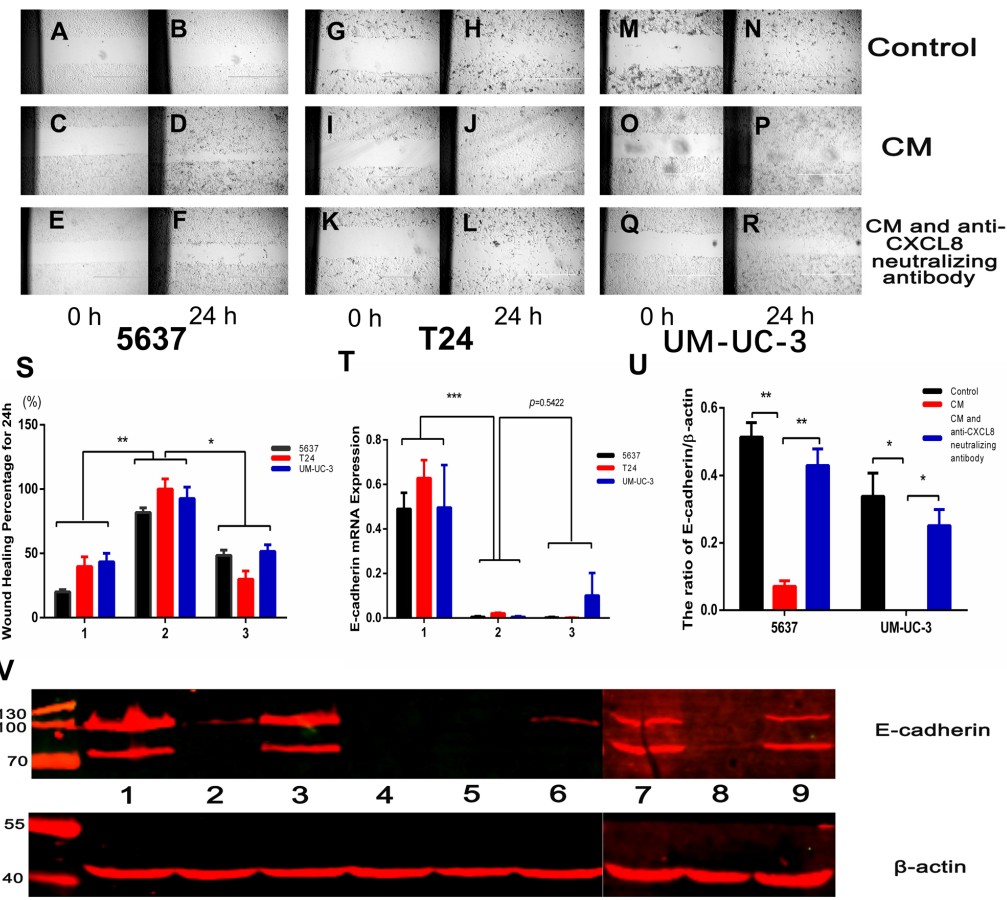

**Figure 4** **AM-like PBM-derived macrophagesincrease the migration of bladder cancer cells through CXCL8. T**(A–R) Three kinds of bladder cancer cell lines (5637, T24, and UM-UC-3) were treated with serum-free RPMI-1640, CM, and CM with anti-CXCL8 neutralizing antibody, respectively. Cell migration was observed at 0, 6, 12, 18, and 24 hours. Pictures taken at 0 and 24 hours. (S) Percentage of wound healing. Formula: wound healing percentage for a certain time (initial area minus area at a certain time point) to the initial area (1, Control; 2, CM; 3, CM with anti-CXCL8 neutralizing antibody). The results were expressed as the mean ± SD. *$p < 0.05$, **$p < 0.01$. (T) The expression level of E-cadherin in 5637, T24, UM-UC-3 was determined by qRT-PCR (1, Control; 2, CM; 3, CM with anti-CXCL8 neutralizing antibody). Expression levels are expressed as the mean ± SD. ***$p < 0.001$. (U–V) 5637, T24, UM-UC-3 cells in serum-free conditions were treated with control, CM and CM with anti-CXCL8 neutralizing antibody. Western blotting was conducted with total protein extracted from bladder cancer cell lines using specific antibodies against E-cadherin and β-actin. The results are mean ± SD. *$p < 0.05$, **$p < 0.01$. (1, 2, 3 were 5637 treated with control, CM and CM with anti-CXCL8 neutralizing antibody; 4,5,6 were T24 treated with control, CM and CM with anti-CXCL8 neutralizing antibody; 7, 8, 9 were UM-UC-3 treated with control, CM and CM with anti-CXCL8 neutralizing antibody).

correlated with the expression of CXCL8 in bladder cancer tissues. This demonstrates that a high expression of CXCL8 is strongly associated with the infiltration of TAMs, and studies have shown similar findings in other cancers (*Fang et al., 2014*; *Lin et al., 2019*).

The CXCL8/CXCL8R axis plays an important role in human cancer and can contribute to tumour progression in multiple ways (*Ha, Debnath & Neamati, 2017*; *Liu et al., 2016*). Under the influence of various stimuli in the tumour microenvironment, stromal cells

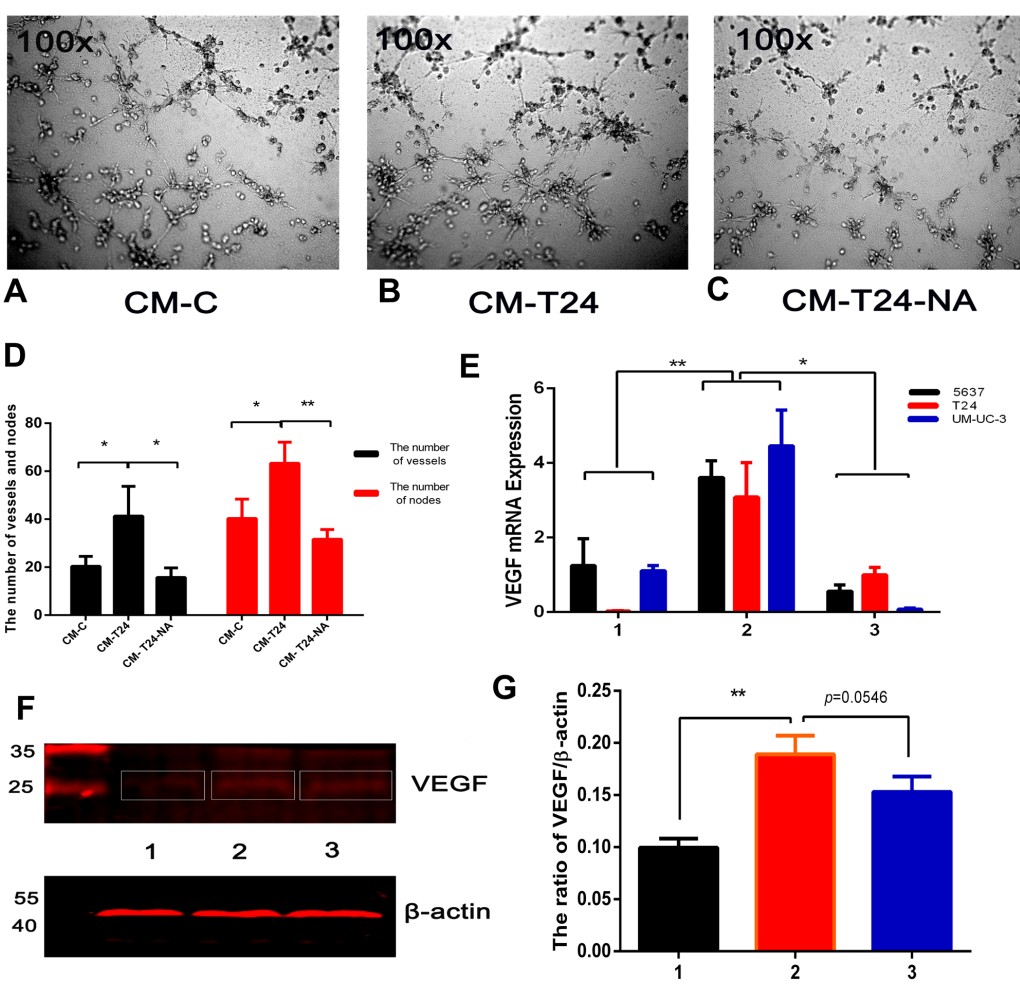

**Figure 5** **CXCL8-derived TAM-like PBM-derived macrophagespromote the pro-angiogenic ability of bladder cancer cells.** (A–C) For endothelial cell tube formation experiments, EA·hy926 cells were added to a Matrigel-coated 96-well. CM-C, CM-T24, and CM-T24-NA were added to wells of a 96-well plate and the tube formation of endothelial cells was observed and photographed. (D) The number of vessels and nodes formed by EA·hy926 cells was counted and plotted on the graph. The results were expressed as the mean ± SD. *$p < 0.05$, **$p < 0.01$. (E) The expression level of VEGF in TAM-like PBM-derived macrophages was determined by qRT-PCR (1, Control; 2, CM; 3, CM with anti-CXCL8 neutralizing antibody). Expression levels are expressed as the mean ± SD. *$p < 0.05$, **$p < 0.01$. (F–G) T24 cells were treated with serum-free medium, CM, and CM with anti-CXCL8 neutralizing antibody. The protein level of VEGF in the bladder cancer cell lines was confirmed by western blotting. Anti-VEGF and β-actin antibodies were used. The results are mean ± SD. *$p < 0.05$, **$p < 0.01$. (1, 2, 3 were T24 treated with control, CM and CM with anti-CXCL8 neutralizing antibody).

produce CXCL8, which may affect the invasiveness and metastatic potential of cancer cells (*Mukaida, 2003*). Blood vessels in tumours provide tumour cells with the necessary nutrients and oxygen, creating convenient conditions for tumour cell growth and invasion (*Cao, 2005*). When stimulated with CXCL8, vascular endothelial cells begin an angiopoiesis process characterized by the secretion of MMP which breaks down the extracellular matrix (*Li et al., 2003*). CXCL8 can significantly increase tumour vessel density in prostate cancer,

which has a very strong angiopoiesis effect (*Araki et al., 2007*; *Inoue et al., 2000a*; *Masuya et al., 2001*). In the present study, we observed that the high expression of CXCL8 in bladder cancer tissues positively correlated with the elevated expression of VEGF, suggesting that CM derived *in vitro* from TAM-like PBM can promote the secretion of VEGF from bladder cancer cells and blood vessel formation. Therefore, the synergistic effect of CXCL8 and VEGF can promote tumour angiogenesis (*Masuya et al., 2001*).

CXCL8-induced MMP-9 promotes extracellular matrix degradation (*Inoue et al., 2000b*), which not only provides the necessary conditions for angiogenesis, but also plays an important role in cancer cell invasion and metastasis. In the present study, a high expression of MMP-9 in cancer tissues was observed by immunohistochemistry, and there was a positive correlation with the increase of CXCL8 expression. Moreover, CM derived from TAM-like PBM-derived CXCL8 could contribute to MMP-9 expression in bladder cancer cells, suggesting that it is also involved in bladder cancer invasion and metastasis. Additionally, when the effect of CXCL8 was inhibited using an anti-CXCL8 neutralizing antibody, we observed that the ability of bladder cancer cells to promote angiogenesis, invasion, and metastasis was significantly reduced.

Epithelial-mesenchymal transition (EMT), a process of transition from epithelial cell phenotype to active mesenchymal cells, plays a key role in tumour progression. TAMs help coordinate this process, which includes the loss of cell–cell contact and the acquisition of a migratory phenotype. The regulation of EMT involves many cytokines and chemokines, including CXCL8 (*Cheng et al., 2014*). IHC assays on clinical specimens in the present study concluded that the expression of E-cadherin decreased under the influence of CXCL8. Moreover, after extracting RNA and protein from bladder cancer cells treated with and without CM, we found that the expression of E-cadherin significantly decreased in bladder cancer cells treated with CM, an effect that could be resisted using an anti-CXCL8 neutralizing antibody (Among bladder cancer cells, T24 has the highest degree of malignancy, which may lead to low or even no expression of E-cadherin in T24). This demonstrates that CXCL8 derived from TAMs can increase bladder cancer cell migration by decreasing the expression of E-cadherin (Figs. 4T–4V).

This study found that infiltrating TAMs can modify the bladder tumor microenvironment by promoting tumor progression with CXCL8. A contrasting study has shown that CXCL8 secreted by tumor cells activates proangiogenic and anti-apoptotic pathways by promoting the expression of VEGFA and bcl-2 at the invasion front, which confirms the key role of CXCL8 in tumor progression (*Kumar et al., 2018*). A similar autocrine mechanism of this CXC chemokine has also been reported (*Bandapalli et al. 2011*; *Matsuo et al., 2009*). By observing immunohistochemical section staining, we found that bladder cancer cells could also secrete CXCL8, meaning that there is complex intercellular communication in the tumor microenvironment involving autocrine and paracrine chemokines. Due to the complexity of the tumor microenvironment and its components, tumor invasion, migration, and angiogenesis may be determined by more than one factor or pathway. Therefore, a macroscopic view of the tumor microenvironment would be effective. This study focused on the role of CXCL8 secreted by TAMs in promoting bladder cancer, but further work is needed to determine the full effect of microenvironment on tumors.

## CONCLUSION

In summary, TAM-derived CXCL8 can promote the expression of MMP-9, VEGF, and E-cadherin in bladder cancer cells, causing changes in bladder cancer cell migration, invasion, and pro-angiogenic ability, and leading to the progression of bladder cancer. The inhibition of CXCL8-signaling derived from TAMs may be a promising therapeutic approach in the treatment of bladder cancer.

### Funding

This study was supported by the Science and Technology project of Chengguan District, Lanzhou city, Gansu province Science and Technology Bureau (Project number: 2017KJGG0052), the Cuiying Graduate Supervisor Applicant Training Program of Lanzhou University Second Hospital (201704), the Gansu Health Industry Research Project (GSWSKY2017-10), the Cuiying Scientific and Technological Innovation Program of Lanzhou University Second Hospital (Project number: CY2017-BJ16, Doctoral supervisor training program), and the National Nature Science Foundation of China (NO: 81372732&30800206). There was no additional external funding received for this study. The funders had no role in study design, data collection and analysis, decision to publish, or preparation of the manuscript.

### Grant Disclosures

The following grant information was disclosed by the authors:
Science and Technology project of Chengguan District, Lanzhou city, Gansu province Science and Technology Bureau: 2017KJGG0052.
Cuiying Graduate Supervisor Applicant Training Program of Lanzhou University Second Hospital: 201704.
Gansu Health Industry Research Project: GSWSKY2017-10.
Cuiying Scientific and Technological Innovation Program of Lanzhou University Second Hospital: CY2017-BJ16.
National Nature Science Foundation of China: 81372732&30800206.

### Competing Interests

The authors declare there are no competing interests.

### Author Contributions

- Hao Wu conceived and designed the experiments, performed the experiments, authored or reviewed drafts of the paper, and approved the final draft.
- Xiangxiang Zhang and Jinlong Cao analyzed the data, prepared figures and/or tables, and approved the final draft.
- Dali Han performed the experiments, analyzed the data, authored or reviewed drafts of the paper, and approved the final draft.
- Junqiang Tian conceived and designed the experiments, authored or reviewed drafts of the paper, and approved the final draft.

## Human Ethics

The following information was supplied relating to ethical approvals (i.e., approving body and any reference numbers):

All procedures performed in studies involving human participants were in accordance with the Ethical Review Committee of the Second Hospital of Lanzhou University. All patients provided written informed consent before the start of the study.

## Data Availability

The raw measurements are provided in the Supplemental Files.

## Supplemental Information

Supplemental information for this article can be found online at http://dx.doi.org/10.7717/peerj.8721#supplemental-information.

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
