# Peer review of "Tumour-associated macrophages mediate the invasion and metastasis of bladder cancer cells through CXCL8"

_PeerJ, doi:10.7717/peerj.8721_

## Round 0.1 · original submission · Major Revisions

Manuscript ID 42475 entitled "Tumour-associated macrophages mediates invasion and metastasis of bladder cancer cells through CXCL" which you submitted to PeerJ, has been reviewed. The reviewers have recommended publication pending major revisions. Therefore, I invite you to respond to the reviewers' comments at the bottom of this letter and revise your manuscript accordingly.

Reviewer 1 ·

Basic reporting

The English language should be improved to ensure that an international audience can clearly understand your text. Some examples where the language could be improved include sentence in introduction and discussion part. The reference, table, and figures are appropriate expect figure 2a.

Experimental design

no comment

Validity of the findings

no comment

Additional comments

This manuscript submitted by Prof. Tian aimed to investigate the effect of TAM on invasion and metastasis in bladder cancer and potential mechanism. After searching the Pubmed, it is the first report on TAM mediates invasion and metastasis through CXCL8. Although it is of some significance for treatment of bladder cancer, but there are some comments:
1. They should analysis relationship between expression of CXCL8 and clinic factors.
2. It is better to show the survival data.

Reviewer 2 ·

Basic reporting

no comment

Experimental design

no comment

Validity of the findings

no comment

Additional comments

This study is novel, the author investigate and clarify the tumour-promoting role of CXCL8 secreted by TAMs in the tumour microenvironment of urothelial carcinoma of the bladder. And find tumour-associated macrophages mediates invasion and metastasis of bladder cancer cells through CXCL8.
Questions
1.Fig1, could the author show us the table, we want to see the expression of these marker in the patients
2.Fig3, the background of this pictures are not the same, and please provide us the westren blot data
3.Fig4 and fig5 please provide us the western blot data

Reviewer 3 ·

Basic reporting

Wu and colleagues in their manuscript titled “Tumour-associated macrophages mediates invasion and metastasis of bladder cancer cells through CXCL8” studied the role of CXCL8 derived from TAMs in the para-epithelial carcinoma network of the human urinary tract in the bladder tumour microenvironment and also analyzed the critical role and biological effects of TAMs-derived CXCL8 on bladder cancer cell production.

Experimental design

The experimental design is good. How did the authors achieve 100x and 400x magnification? The clarity of the figures is not good.

Validity of the findings

The findings are good and in line with the known roles of CXCL8 in other cancers. Please discuss these results based on the paper “ShRNA-mediated knock-down of CXCL8 inhibits tumor growth in colorectal liver metastasis” where the authors showed that CXCL8 secreted by tumor cells at the invasion front were able to promote migration through angiogenesis by upregulating VEGFA and invasion via the AKT/GSK3β/β-catenin/MMP7 pathway by upregulating BCL-2 confirming the key role of CXCL8 during tumor progression

Additional comments

The abstract needs to be rewritten briefly. It looks like material and methods section.

---

## Round 0.2 · accepted · Accept

Thanks for the revision of the manuscript, which can be now accepted.

Reviewer 1 ·

Basic reporting

No comments

Experimental design

No comments

Validity of the findings

No comments

Additional comments

The authors have changed the manuscript according to the comments, I think it is acceptalbe in its current version.

Reviewer 2 ·

Basic reporting

N/A

Experimental design

N/A

Validity of the findings

N/A

Additional comments

The authors have answered all my questions.

Reviewer 3 ·

Basic reporting

Th authors have answered all my queries and the manuscript has improved and no further comments from me.

Experimental design

Th authors have answered all my queries and the manuscript has improved and no further comments from me.

Validity of the findings

Th authors have answered all my queries and the manuscript has improved and no further comments from me.

Additional comments

Th authors have answered all my queries and the manuscript has improved and no further comments from me.